# MG53 Mitigates Nitrogen Mustard-Induced Skin Injury

**DOI:** 10.3390/cells12141915

**Published:** 2023-07-23

**Authors:** Haichang Li, Zhongguang Li, Xiuchun Li, Chuanxi Cai, Serena Li Zhao, Robert E. Merritt, Xinyu Zhou, Tao Tan, Valerie Bergdall, Jianjie Ma

**Affiliations:** 1Department of Veterinary Biosciences, The Ohio State University, Columbus, OH 43210, USA; 2Department of Surgery, The Ohio State University, Columbus, OH 43210, USA; zhongguanglee@sina.com (Z.L.); lxchosu@hotmail.com (X.L.); chuanxi.cai@virginia.edu (C.C.); serenali.zhao@osumc.edu (S.L.Z.); robert.merritt@osumc.edu (R.E.M.); xinyu.zhou@virginia.edu (X.Z.); 3TRIM-Edicine, Inc., 1275 Kinnear Road, Columbus, OH 43212, USA; ttan@trim-edicine.com; 4Department of Veterinary Preventive Medicine, The Ohio State University, Columbus, OH 43210, USA; bergdall.1@osu.edu

**Keywords:** alkylating agents, MG53, membrane repair, dermal healing, oxidative stress

## Abstract

Sulfur mustard (SM) and nitrogen mustard (NM) are vesicant agents that cause skin injury and blistering through complicated cellular events, involving DNA damage, free radical formation, and lipid peroxidation. The development of therapeutic approaches targeting the multi-cellular process of tissue injury repair can potentially provide effective countermeasures to combat vesicant-induced dermal lesions. MG53 is a vital component of cell membrane repair. Previous studies have demonstrated that topical application of recombinant human MG53 (rhMG53) protein has the potential to promote wound healing. In this study, we further investigate the role of MG53 in NM-induced skin injury. Compared with *wild-type* mice, *mg53^−/−^* mice are more susceptible to NM-induced dermal injuries, whereas mice with sustained elevation of MG53 in circulation are resistant to dermal exposure of NM. Exposure of keratinocytes and human follicle stem cells to NM causes elevation of oxidative stress and intracellular aggregation of MG53, thus compromising MG53′s intrinsic cell membrane repair function. Topical rhMG53 application mitigates NM-induced dermal injury in mice. Histologic examination reveals the therapeutic benefits of rhMG53 are associated with the preservation of epidermal integrity and hair follicle structure in mice with dermal NM exposure. Overall, these findings identify MG53 as a potential therapeutic agent to mitigate vesicant-induced skin injuries.

## 1. Introduction

Sulfur mustard (SM) is an alkylating agent that can cause painful blistering of the skin, as well as harm to the eyes and respiratory tract [1,2,3,4,5,6,7]. It is a readily available and inexpensive substance that can be stockpiled, making it a potential weapon that could pose a significant threat to both civilians and military personnel. Because SM is highly regulated and requires specialized containment facilities, studying and developing effective treatments for SM-induced tissue injuries have been challenging. Many studies have shown that nitrogen mustard (NM), an analog of SM, induces comparable injuries in humans and other animal models. Thus, NM has been widely used as a substitute in laboratory settings to investigate the mechanism and develop novel therapeutic [8,9,10,11,12,13,14,15].

The toxic effects of mustard vesicants on the skin primarily stem from their ability to damage DNA, generate free radicals, and trigger lipid peroxidation. These injury processes are associated with the alkylating properties and/or thiol-depleting functions of SM and NM [5,9,16,17,18,19,20,21]. Oxidative stress and cell membrane damage caused by SM and NM are major factors contributing to the development of debilitating conditions such as blistering, chronic inflammation, and ulceration [16,19,20,22,23,24]. Due to their lipophilic nature, SM/NM can penetrate deep into the epidermal layer and cause damage to keratinocytes and hair follicles [25]. While vesicant-induced damage to keratinocytes and hair follicle stem cells (HFSCs) contribute to the acute phase of cutaneous injury, sustained inflammation, and loss of HFSCs are the underlying cause for the delayed phase of ulceration and penetrating damage to the skin [26,27].

An emerging concept in recent biomedical research establishes that intrinsic membrane repair/regeneration is a fundamental aspect of normal human physiology and that disruption of this repair function underlies the progression of several pathologies [28,29,30,31,32], including wound healing, chronic ulcers, and scarring. Preserving the function of keratinocytes and HFSCs could be a promising therapeutic strategy to counteract vesicant-induced skin lesions.

Previously, we identified a novel TRIM family protein, named MG53, as an essential component of the cell membrane repair machinery [33,34]. Mice with ablation of the MG53 gene (*mg53^−/−^*) develop pathology in multiple tissues and organs due to defective cell membrane repair [35,36,37]. The recombinant human MG53 (rhMG53) protein can protect various cell types against membrane disruption and ameliorate pathology associated with muscular dystrophy [38], myocardial infarction [39], acute kidney injury [40], and lung injury [41].

We have shown that MG53 is a vital component of wound healing and that topical application of the rhMG53 protein has the potential to promote wound healing and suppress scar formation [37,42,43]. As a facilitator to protect against injuries to the epidermis, MG53 can enhance reepithelization and reduce the overall inflammatory responses during the early phase of wounding. Moreover, MG53 plays an important role in the rejuvenation of hair follicle stem cell (HFSC) function, and sustained delivery of rhMG53 promotes diabetic wound healing [44].

In this study, we present compelling evidence supporting the physiological role of MG53 in protection against mustard-induced skin injuries. We found that exposure of HFSCs to NM caused intracellular aggregation of MG53 which compromises MG53′s intrinsic cell membrane repair function. rhMG53 applied topically mitigates NM-induced dermal injury, through reduction of cellular oxidative stress, preservation of epidermal layer integrity, and HFSC function.

## 2. Materials and Methods

### 2.1. Regents, and Recombinant Human MG53 Protein (rhMG53)

NM (Mechlorethamine hydrochloride) was purchased from Sigma-Aldrich Chemicals Co. (St. Louis, MO, USA). rhMG53 protein was produced as described previously ^38^.

### 2.2. Animals and NM Exposure

All animal care and usage followed NIH guidelines and IACUC approval by The Ohio State University. Eight to ten-week-old male *mg53^−/−^* mice, tPA-MG53 mice (transgenic mice with sustained elevation of MG53 in the bloodstream [43], and their *wild-type* littermates were bred and generated as previously described [33,43]. Age-matched C57BL/6 mice were purchased from Charles River Laboratories. Mice were anesthetized using isoflurane. Dorsal hair was shaved using electric clippers and cleared by *Nair^TM^* (hair remover lotion, Church and Dwight Co., Inc., Princeton, NJ, USA) 3 days before NM exposure. Mice were randomly assigned, and dorsal skin was topically exposed with NM in 50 µL acetone/mouse or 50 µL acetone alone as control following the protocol as described previously with minor modification [19,20]. The NM doses were selected based on the previous studies [5,19,20,45]. 3.2 mg NM/mice were used for the systemic toxicity study, and 2.0 mg NM/ mice and 0.8 mg NM/mice were used for the acute and delayed injury study. NM was prepared immediately before administration. All procedures were performed in a designated room with a chemical hood strictly following OSU Environment and Health and Safety guidelines. rhMG53 (2 mg/kg) in saline or saline only was administered by daily subcutaneous (SC) injection for 5 days post NM exposure.

### 2.3. Cells, Cell Culture, and Stress Treatment

Human hair follicle stem (HFSC) cells were purchased from Celprogen Inc. (Torrance, CA, USA). Human keratinocytes were purchased and cultured as described previously [37]. The cells were grown in RPMI 1640 medium supplemented with 10% FBS, 100 U/mL penicillin, and 100 μg/mL streptomycin at 37 °C in the presence of 5% CO_2_. HFSC cells used were 3–8 passages cultured in the complete media.

### 2.4. Apoptosis Assay

Cell apoptosis was investigated by dual staining with Alexa Fluor 488-Annexin V and Propidium Iodide (PI) (Invitrogen Cat# V13241) following the manufacture protocol and analyzed as described previously [46,47]. Briefly, HFSC cells were seeded in 6-well plates and cultured for 24 h, then incubated with 50 µM NM or BSA for 4 h and incubated with vehicle or rhMG53 (10 µg/mL) for another 20 h. Cells were detached by 0.25% Trypsin-EDTA solution. Annexin V & PI staining were performed for FACS analysis. The intensity of fluorescence was detected by Guava EasyCyte™ System and analyzed by GuavaSoft™ Module software (GuavaSoft™ 3.3, EMD Millipore Corporation, Inc. St. Louis, MO, USA). The four quadrants (Q1, Q2, Q3, and Q4) are marked in all FASC analyses. Cells positive for Annexin V only (Q4), or double positive for both Annexin V and PI (Q2) were defined as apoptotic cells (Q2 + Q4); cells positive for PI only (Q1) were defined as necrotic cells; and cells negative for both Annexin V and PI (Q3) were defined as live cells.

### 2.5. ROS Measurement

Cellular reactive oxygen species (ROS) production was measured using a Cellular ROS Detection Assay Kit (Abcam Cat#ab113851) according to manufacturer instructions and analyzed as described previously [46,47]. Briefly, HFSC cells were seeded in 6 well plates and cultured for 24 h, then incubated with 50 µM NM or BSA for 4 h, and incubated vehicle or rhMG53 (10 µg/mL) cultured for another 20 h. Cells were washed with PBS and re-suspended with 100 μL assay buffer including 1× ROS Deep Red Dye and incubated in 5% CO_2_, 37 °C for 30 min. The intensity of red fluorescence was detected by Guava EasyCyte™ System and confocal images were taken by confocal microscope.

### 2.6. Cell Membrane Injury Assay and Confocal Microscopy

For membrane repair assay, HFSC cells were transfected with GFP-MG53, then subjected to microelectrode penetration-induced acute injury to the plasma membrane, and the data were analyzed as previously described [33].

### 2.7. Histology

Histology staining was performed as previously described [37]. Briefly, skin tissues were dissected from experimental animals and then fixed in 4% Paraformaldehyde (PFA) overnight at 4 °C. After fixing, samples were washed three times for 5 min with 70% ethanol. Washed samples were processed, embedded in paraffin, and 4-μm thick paraffin sections were cut. Cells were fixed with 4% PFA.

### 2.8. Statistical Analysis

All data are expressed as means ± standard error of the mean (SEM). For each experiment, three independent replicates were performed. Statistical evaluation was conducted using the Student’s *t*-test and one-way ANOVA. A value of *p* < 0.05 was considered statistically significant.

## 3. Results

### 3.1. Dermal Exposure of NM Causes Severe Skin and Systemic Injury in mg53^−/−^ Mice

We first compared the response of the *wild-type* and *mg53^−/−^* mice (8–10 weeks age, male) to dermal exposure of NM. When NM (3.2 mg/mouse) was applied to the dorsal skin, the *mg53^−/−^* mice developed a more severe response at day 2 post-exposure and became less active compared to the *wild-type* littermates (Figure 1a, see also Appendix A) due to systemic toxicity. We examined the morphology changes in organ weight and intestinal length between *mg53^−/−^* and age-matched *wild-type* mice and did not observe any significant difference with those phenotypes. At day 5 post NM exposure, mice were sacrificed, and histological examinations were conducted with the major vital organs (including heart, lung, liver, kidney, spleen, and colon) (Figure 1b,c). Cutaneous exposure to NM caused measurable shrinkage in the spleen and liver, to a greater extent in the *mg53^−/−^* mice than *wild-type* mice (Figure 1e). Increased reduction in the colon length derived from *mg53^−/−^* mice compared to *wild-type* mice reflects intestinal toxicity caused by NM exposure (Figure 1c). Histological evaluations revealed more severe tissue injuries in the liver, spleen, and colon derived from *mg53^−/−^* mice compared with *wild-type* mice (Figure 1d). These studies support the physiological role of MG53 as a tissue protector, and ablation of MG53 renders the mice more susceptible to NM-induced systemic toxicity.

We next conducted experiments using a lower dose of NM cutaneous exposure (0.8 mg/mouse). Compared with *wild-type* mice, exacerbated blistering was observed at day 2 and day 9 post NM exposure (0.8 mg/mouse) in the *mg53^−/−^* skin (Figure 2a). We demonstrated previously that mice with sustained elevation of MG53 in circulation (tPA-MG53) have increased regenerative capacity following tissue injury [43]. As shown in Figure 2a, tPA-MG53 mice displayed less severe skin injury following NM exposure. Histological analyses revealed that the hair follicle development was severely compromised in *mg53^−/−^* skin, compared with those in *wild-type* and tPA-MG53 mice at 28 days post NM exposure (Figure 2b).

### 3.2. rhMG53 Treatment Mitigates NM-Induced Epidermal Structure Damages

In addition to the topical application of rhMG53, we have previously demonstrated that intramuscular (IM) or subcutaneous (SC) administration of rhMG53 had therapeutic benefits to treat dermal wounds [37]. Here we conducted studies to evaluate the therapeutic benefit of rhMG53 applied both topically and SC in treating NM-induced cutaneous injury in C57BL/6J mice (8 weeks age, male). The dorsal skin of the C57BL/6J mice was shaved and then exposed to NM (2 mg/mouse). Topical applications of the rhMG53/cream (3 mg/oz) plus SC injections (1 mg/kg, near the wound site) were applied daily for a total of 3 days. Control mice received topical treatment with cream alone and SC injection of saline (equal volume of 50 µL as rhMG53).

As shown in Figure 2d, following 3 days of treatment with rhMG53, a visible difference in skin architecture could be observed. We quantified the changes in thickness of the epidermal layer caused by NM exposure based on H/E staining of the affected skin area and found a significant reduction in mice treated with rhMG53 compared with those treated with saline control (Figure 2c).

In a recent study [44], we showed that mice with ablation of MG53 display defective hair follicle structure, and topical application of rhMG53 promoted hair growth in the *mg53^−/−^* mice. It will be interesting to study more on hair follicle stem cell function and the long-term effect of rhMG53 treatment including the hair follicle development following NM exposure in the future.

### 3.3. NM-Induced Oxidative Stress Impacts MG53′s Intrinsic Membrane Repair Function in HFSC

Toward understanding the mechanism that underlies MG53′s role in protection against NM-induced skin injury, we conducted in vitro studies with cultured HFSCs. HFSCs were transfected with GFP-MG53. At 24 h after transfection, confocal fluorescence live cell imaging was conducted with control HFSCs and those treated with 50 µM NM. As shown in Figure 3a (top), there were remarkable changes in the subcellular distribution of GFP-MG53 in HFSCs treated with NM. GFP-MG53 shows a predominant plasma membrane localization and cytosolic distribution in control conditions, whereas cells treated with NM display aggregated GFP-MG53 in the cytosol (Figure 3a, top). We used microelectrode poking to induce acute injury to the plasma membrane of HFSCs and observed rapid translocation and accumulation of GFP-MG53 toward the acute plasma membrane injury site, which was expected for the membrane repair-patch function of MG53 (as observed in many other cell types [33,37,40,41] (Figure 3b, top). The repair patch remained stable over the 2 min observation period in control HFSCs that were not exposed to NM (blue). Remarkably, HFSCs treated with NM displayed dysfunctional movement of GFP-MG53 following microelectrode poking. While there was a small initial accumulation of GFP-MG53 at the injury site within the first 30 s after injury, the repair patch did not remain stable as they decreased within the 2 min recording period (red).

Studies from other investigators have shown that dermal exposure to SM/NM leads to massive elevation of cellular oxidative stress [13,16,18,48,49,50,51]. We have demonstrated before that MG53′s intrinsic membrane-repair function is impaired when cells are subjected to chronic oxidative stress [52,53,54,55]. Thus, these findings provide a novel mechanism for NM-induced tissue injury that involves oxidative-stress-mediated disruption of the endogenous cell membrane repair machinery. It also lays the foundation for the use of exogenous rhMG53 to boost the defense mechanism of the cells associated with vesicant-induced tissue injuries.

### 3.4. rhMG53 Protects against NM-Induced Injury to HFSCs and Keratinocytes

We next conducted studies to investigate the potential protective role of rhMG53 in mitigating NM-induced oxidative stress and injury to HFSCs and keratinocytes. We used DCF fluorescent indicator to quantify the intracellular reactive oxygen species (ROS) level and observed a significant increase in DCF fluorescence in HFSCs treated with NM (50 µM, for 2 h). rhMG53 treatment (5 µg/mL) suppressed NM-induced elevation of ROS in HFSCs (Figure 4a). The summary data from multiple experiments demonstrated that rhMG53 could mitigate NM-induced oxidative stress in HFSCs (Figure 4c). A similar finding was observed in keratinocytes (Figure 4d).

A 2′,7′-dichlorodihydrofluorescein diacetate (DCFH-DA,) staining is used for quantification of ROS detection [56]. Oxidation of DCFH by ROS converts the molecule to DCF, which emits green fluorescence at an excitation wavelength of 485 nm and an emission wavelength of 530 nm. FACS analyses were thus used to quantify the changes in DCF levels in HFSCs treated with NM (+saline, pink) and NM (+rhMG53, green) (Figure 4b). Clearly, the left shift in the peak distribution of DCF reflected the anti-oxidative stress function of rhMG53 in HFSCs following NM exposure.

To quantify the degree of NM-induced cell death, we next conducted FACS analysis with the labeling of propidium iodide and Annexin V (Figure 5a). Clearly, NM-induced cell death was reduced by rhMG53 treatment. Summary data from multiple experiments are shown in Figure 5b. Together, these findings support the notion that NM-induced oxidative stress is protected by MG53, which may underlie the dermal protective function of rhMG53 following exposure to NM.

## 4. Discussion

Our findings indicate that *mg53^−/−^* mice are more susceptible to NM-induced dermal injuries, while tPA-MG53 mice (transgenic mice with sustained elevation of MG53 in the bloodstream) show resistance to NM exposure. The application of exogenous rhMG53 can mitigate NM-induced cellular oxidative stress, and effectively mitigate NM-induced dermal injury in mice, with preserved epidermal integrity and hair follicle structure. Thus, targeting cell membrane injury repair could serve as a potentially effective approach to protect skin from damages caused by mustard vesicants.

Accumulating evidence suggests that alkylating agents, SM/NM, cause skin injuries through complicated cellular events, involving DNA damage, free radical formation, and lipid peroxidation [5,9,10,13,16,17,18,20,24,57,58,59]. We found that exposure of keratinocytes and HFSCs to NM leads to the elevation of oxidative stress. rhMG53 treatment can improve the survival of the cells and reduce ROS levels upon exposure to NM. In cells treated with NM, GFP-MG53 is incapable of translocation to the membrane injury site for repair-patch formation. These findings are consistent with our previous study that reveals a role for oxidative stress in the control of MG53′s membrane repair function [54,55]. Similar findings were also observed with lung epithelial and endothelial cells [46]. Thus, NM-induced tissue injury and oxidative stress can all be mitigated by rhMG53, laying the foundation for the use of exogenous rhMG53 to boost the defense mechanism against vesicant-induced tissue injury.

Previous studies by Au et al. have shown that 25-hydroxyvitamin D3 (25(OH)D) has efficacy in improving the healing process of the skin by modulating oxidative stress and inflammation following vesicant exposure [22]. Our study demonstrated that rhMG53 has both tissue-repair and anti-inflammation functions associated with the healing process of the skin following NM exposure. In principle, the dual function of MG53 would have an advantage over 25(OH)D or other reagents that are currently being developed for wound healing applications. Clearly, more studies are needed to advance the pre-clinical findings with rhMG53 into human applications.

Given that MG53 is present in circulation under normal physiologic conditions, topical or IM administration of rhMG53 are not likely to produce neutralizing antibodies as peripheral tolerance to this protein has already been established [38]. Pharmacokinetic and toxicology assessment also supports the safety of repetitive IV administration of rhMG53 in rodents and large animals [40]. Thus, a therapeutic approach that modulates endogenous MG53 levels/function or involves systemic administration of rhMG53 protein is potentially a safe biologic means to treat and prevent tissue damage, including vesicant-induced multi-organ injury.

In a recent study [44], we showed that mice with ablation of MG53 display defective hair follicle structure, and topical application of rhMG53 can promote hair growth in the *mg53^−/−^* mice. We further find that rhMG53 protects HFSCs from oxidative stress-induced apoptosis and stimulates the differentiation of HSFCs into keratinocytes [44]. Additionally, MG53 has been shown to play an anti-inflammatory role in viral infection, chronic injury, and aging [60,61]. All these findings link MG53′s function to the improved regenerative capacity to facilitate wound healing. It remains to be established how the dual function of MG53 in tissue repair and anti-inflammation is linked to the potential long-term benefits of rhMG53 as a potential therapeutic to treat vesicant-induced dermal injury.

## 5. Conclusions

Overall, this study demonstrates a physiologic role for MG53 in protection against vesicant-induced dermal injury. We provide evidence that topical rhMG53 application preserves epidermal integrity and hair follicle structure following NM exposure. Future studies are required to examine if sustained rhMG53 administration can mitigate the inflammatory component of dermal injury associated with vesicant exposure.

## Figures and Tables

**Figure 1 cells-12-01915-f001:**
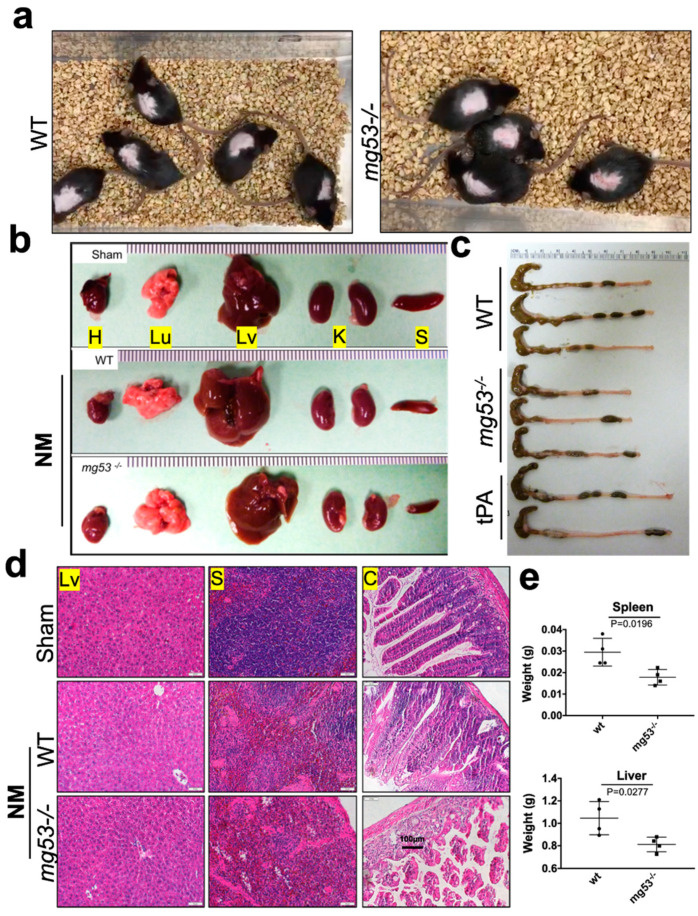
High dose of NM exposure on the dorsal skin causes systemic injury in *mg53^−/−^* mice. Dorsal skin of mice was exposed to NM (3.2 mg/mouse) in 50 µL acetone or 50 µL acetone alone. (**a**) Representative images of *wild-type* (left panels) and *mg53^−/−^* (right panels) mice at day 2 post-NM exposure. (**b**) Left panel, representative images of multiple organs from *wild-type* sham, and *wild-type* and *mg53^−/−^* mice at day 5 post-NM exposure (3.2 mg/mouse). (**c**) Representative images of colon from *wild-type*, *mg53^−/−^* and tPA mice at day 28 post-NM exposure (2.0 mg/mouse). (**d**) H&E images of liver (left panel), spleen (middle panel) and colon (right panel) from *wild-type* sham, wild-type, *mg53^−/−^* at day 5 post-NM exposure (3.2 mg/mouse). (**e**) Qualification of organ weight for spleen and liver at day 5 post-NM exposure (3.2 mg/mouse) (n = 8, per group).

**Figure 2 cells-12-01915-f002:**
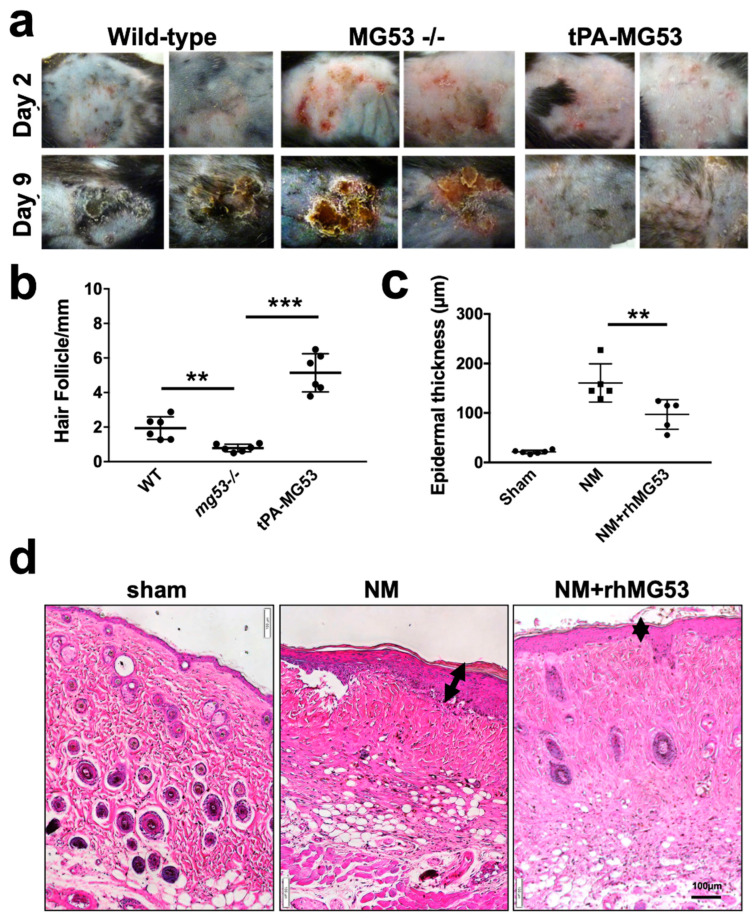
NM exposure on the dorsal skin causes severe skin injury in *mg53^−/−^* mice and rhMG53 treatment mitigates skin injury following NM exposure. Dorsal skin of mice was exposed to NM (0.8 mg/mouse) in 50 acetone or 50 µL acetone alone. (**a**) Representative images of dorsal skin of *wild-type* (left panels) and *mg53^−/−^* (middle panels), tPA-MG53 (right panels) mice at indicated time points post-NM exposure. (**b**) Quantification of hair follicles at 28 days post NM exposure (0.8 mg/mice, n = 6, per group). (**c**) Quantification of the epidermal thickness at the day 3 post NM exposure (2.0 mg/mouse) (n = 5, per group). (**d**) Dorsal skin of mice was exposed to NM (2.0 mg/mouse). Representative H&E images from saline control (left), NM (middle), and rhMG53 treatment (right) of skin at the day 3 post NM exposure. ** *p* < 0.01, *** *p* < 0.001 for the indicated group. Dot represents an individual mouse.

**Figure 3 cells-12-01915-f003:**
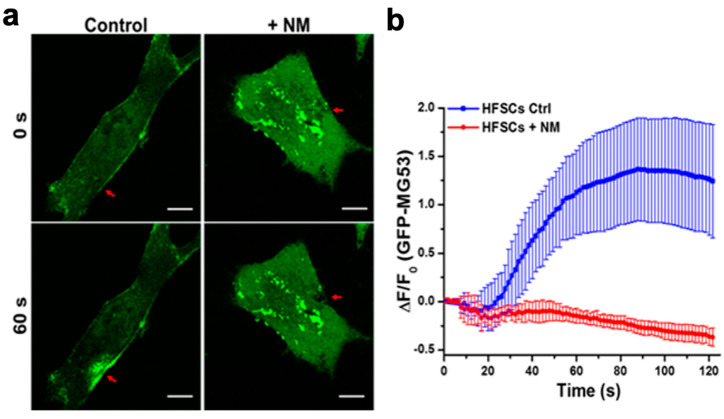
NM exposure interrupts MG53-medicated membrane repair in HFSCs. (**a**) HFSC cells were transfected with GFP-MG53. At 24 h after transfection, confocal fluorescence live cell imaging was conducted with control HFSC cells and those treated with 50 µM NM, which were injured by penetration of a microelectrode. Upper panel—cell image taken immediately after injury; lower panel—image taken 60 s after NW treatment. Arrows show the microelectrode injury site. Scale bar, 5 µm. (**b**) Time-course of GFP-MG53 accumulation at the injury sites following microelectrode penetration in NM-treated HFSC cells. The repair patch remained stable over the 2 min observation period in control cells not exposed to NM (green). Cells treated with NM displayed dysfunctional GFP-MG53 movement following microelectrode-induced membrane injury (red).

**Figure 4 cells-12-01915-f004:**
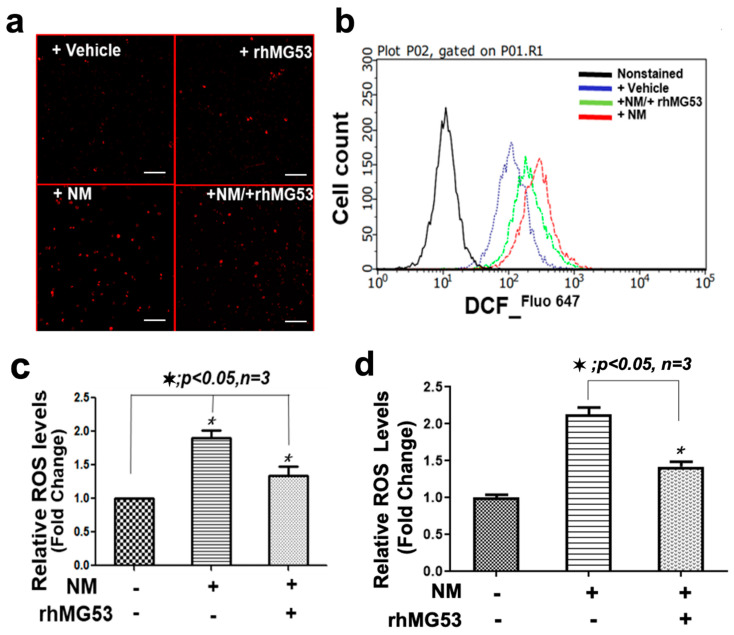
rhMG53 attenuates NM-induced oxidative stress in keratinocytes and HFSC cells. HFSCs and keratinocytes were treated with 50 µM NM or vehicle for 4 h and washed and incubated vehicle or rhMG53 (10 µg/mL) cultured for 20 h. (**a**) Representative image for HFSCs stained with a Cellular ROS detection kit. Scale bar, 25 µm. (**b**) Representative FACS analyses of ROS levels with DCF staining for HFSCs were presented with non-stained cells (black), vehicle-treated cells without NM (blue), NM-treated cell + saline (pink), and NM-treated cells + rhMG53 (green). (**c**) Quantification of relative ROS levels for HFSCs based on the Cellular ROS detection kit (*n* = 3). (**d**) Quantification of relative ROS levels for *keratinocytes* based on the Cellular ROS detection kit (*n* = 3).

**Figure 5 cells-12-01915-f005:**
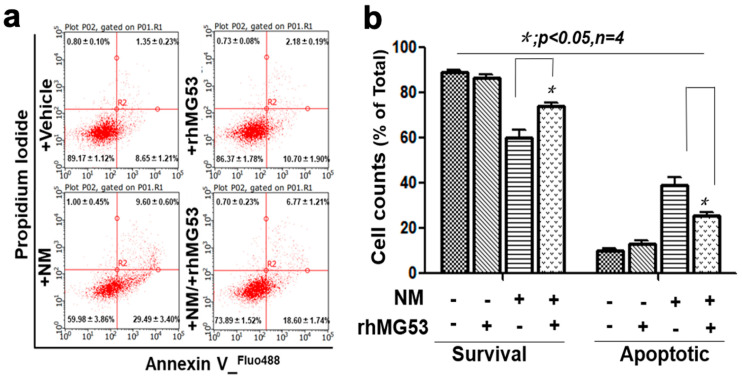
rhMG53 protects HFSC cells from NM-induced cell death. The HFSC cells were cultured for 24 h, then treated with 50 µM NM or BSA for 4 h, and then incubated with rhMG53 (10 µg/mL) or BSA control cultured for another 20 h. Cell was stained and analyzed with Annexin V & PI staining. (**a**) Representative apoptosis analysis of HFSCs (FACS). (**b**) Quantification of HFSC apoptosis for panel (**a**) (*n* = 4).

## Data Availability

The datasets for this study are available from the corresponding author on reasonable requests.

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
