# Peer review of "MG53 Mitigates Nitrogen Mustard-Induced Skin Injury"

_cells, 2023, doi:10.3390/cells12141915_

Round 1
Reviewer 1 Report
This research is an important contribution to the field of vesicant research since approved therapeutic agents to alleviate vesicant-induced skin injuries are still needed. The authors report that topical recombinant human MG53 (rhMG53) , with potential to promote wound healing and reduce scar formation, mitigates nitrogen mustard (NM)-induced dermal injury in mice. The study was conducted using wild type mice, mg53-/- mice ad mice with sustained elevation of MG53 in circulation.
There are several comments and concerns that need attention:
1. Why were the reported NM concentrations used and how these relate to sulfur mustard (SM) vapor exposure? This needs to be added in the discussion section. The different concentrations and time points of study need to be included in the methods section.
2. In the methods indicate that how long after the mice were exposed to NM after shaving.
3. It will be useful to mention the relevance of the study at different time points and what is the relevance of these with SM exposure and its injury progression.
4. It will be helpful if some detail or what are the tPA-MG53 mice-is introduced to the readers.
5. It is hard to see the histopathology in Figure 2b. A better resolution and size will be better.
6. The authors here report that exogenous 254 rhMG53 can mitigate NM-induced cellular oxidative stress. Since they have done in vivo mice studies with KO and overexpressing mice, it will be relevant to show some oxidative damage markers to confirm this in in vivo studies.
7. No molecular inflammatory or DNA damage marker studies in in vivo experiments were carried out.
8. The discussion does not include how the MG53 treatment will be better as compared to other treatments used in the reported studies.
Reviewer 2 Report
Li et al have convincingly showed the role of MG53 as a mitigator of mustard induced skin injuries. The authors must reduce the number of references cited. there are few typos in the text eg line 237.
there are few typos in the text eg line 237.
Author Response
Li et al have convincingly showed the role of MG53 as a mitigator of mustard induced skin injuries. The authors must reduce the number of references cited. there are few typos in the text eg line 237.
there are few typos in the text eg. line 237.
Response:
We appreciate the reviewer’s endorsement and have corrected typos in revised manuscript.

Reviewer 3 Report
This study examines the mitigation of NM-induced skin injury by MG53, a protein involved in cell membrane repair. Overall, the manuscript is clear and well-written, and the data are supportive of the conclusions. Some controls and experimental details are missing. See comments below.
1. The introduction would be aided by including more mechanistic details on what is known about how MG53 functions in cell membrane repair.
2. Include all doses of NM exposure in the Materials and Methods section.
3. Please include age of mice in Materials and Methods section. Were mice age-matched?
4. Clarify fixation of tissue samples. Were the samples fixed twice? (lines 121, 123)
5. Please include which post-test was used for statistical analyses by ANOVA.
6. Figure 1.
a. (Fig 1a) Line 134 indicates a more severe cutaneous response to NM in the null mice. It is difficult to visualize this with images/videos provided. A magnified representative image of the treated skin area for each genotype would be helpful.
b. (Fig 1b) Either label the organs in the figure or indicate it in the legend.
c. (Fig 1d) It is difficult to visualize liver injuries as stated. Perhaps including a magnified panel or outlining areas of damage would be helpful. The colon image for the null mouse appears to be in a different plane of section, making it difficult to appreciate histological differences. Please clarify.
d. (Fig 1c and e) Since WT and mg53 -/- mice are different genotypes, it is necessary to analyze organ weight/length in untreated, age-matched animals to conclude the differences observed are due to NM treatment response and not, in part, due to normal physiological differences.
7. Figure 2.
a. Include data for acetone (vehicle) controls (Fig 2 a-c)/control mice with topical treatment alone and SC injection of saline (Fig 2 d-e) here or in supplemental data.
b. (Fig 2c) See comment 5d. Same should be done for hair follicle quantitation.
c. Line 172 indicates images are from lung. Revise.
d. If possible, it is recommended to make Fig 2d-e a separate, additional figure since they are separate from Fig 2a-c experimentally.
e. (Fig 2d) It is unclear why hair follicles were not quantified here.
8. Line 177, section title is unclear as written.
Author Response
Reviewer 3:
Comments and Suggestions for Authors
This study examines the mitigation of NM-induced skin injury by MG53, a protein involved in cell membrane repair. Overall, the manuscript is clear and well-written, and the data are supportive of the conclusions. Some controls and experimental details are missing. See comments below.
- The introduction would be aided by including more mechanistic details on what is known about how MG53 functions in cell membrane repair.
We agree with the reviewer and have revised the Introduction accordingly to add more background information about MG53 in cell membrane repair and tissue regeneration. See line 71-82.
- Include all doses of NM exposure in the Materials and Methods section.
Agree. Please also see our response to reviewer 1.
- Please include age of mice in Materials and Methods section. Were mice age-matched?
Yes, we used age-matched wild type mice for our comparative studies. The details have been added to the Materials and Methods section.
- Clarify fixation of tissue samples. Were the samples fixed twice? (lines 121, 123)
In our studies, all samples were fixed in 4% once.
- Please include which post-test was used for statistical analyses by ANOVA.
We have provided the detailed statistical analysis on page 5, line 151-152.
- Figure 1.
- (Fig 1a) Line 134 indicates a more severe cutaneous response to NM in the null mice. It is difficult to visualize this with images/videos provided. A magnified representative image of the treated skin area for each genotype would be helpful.
Sorry for the low magnification. We have modified Fig.1 accordingly. We mentioned the more severe systemic injury following NM cutaneous exposure based on the observation of animal behavior (Fig 1a). Please see the Supplementary movies for details.
- (Fig 1b) Either label the organs in the figure or indicate it in the legend.
We have following the reviewer’s suggestion and modified Fig. 1 as recommended.
- (Fig 1d) It is difficult to visualize liver injuries as stated. Perhaps including a magnified panel or outlining areas of damage would be helpful. The colon image for the null mouse appears to be in a different plane of section, making it difficult to appreciate histological differences. Please clarify.
Thanks for your suggestion.
In this study, we showed that topical NM application caused the reduced liver weight without significant pathological changes as shown in 1d, which is consistent with the previous studies by Goswami (2015). The morphological images of the colon were from mice at day 28 following topical NM exposure (2.0 mg/mouse). The histological images of the colon were from the mice at the day 5 following NM topical NM exposure (3.2 mg/mouse).
- (Fig 1c and e) Since WT and mg53 -/- mice are different genotypes, it is necessary to analyze organ weight/length in untreated, age-matched animals to conclude the differences observed are due to NM treatment response and not, in part, due to normal physiological differences.
We have analyzed all organ-weight in different genetic ground following NM exposure (2 mg/mouse). We used sham as untreated control. The significant differences were only observed in liver and spleen.
- Figure 2.
- Include data for acetone (vehicle) controls (Fig 2 a-c)/control mice with topical treatment alone and SC injection of saline (Fig 2 d-e) here or in supplemental data.
These details have been added to the legend for Fig. 2. Please see line 310-315.
- (Fig 2c) See comment 5d. Same should be done for hair follicle quantitation.
The hair follicle was quantified at the day 28 following NM exposure (as shown in Fig. 2d).
- Line 172 indicates images are from lung. Revise.
Sorry for the typo mistake. We have changed “lung” to “skin”.
- If possible, it is recommended to make Fig 2d-e a separate, additional figure since they are separate from Fig 2a-c experimentally.
To save space, we have put MG53-/- and rhMG53 treatment data together to show that MG53 mitigates skin injury following NM exposure.
- (Fig 2d) It is unclear why hair follicles were not quantified here.
The hair follicle was quantified at the day 28 following NM exposure (Fig. 2d).
Line 177, section title is unclear as written.
Thanks for your suggestion. We have revised sections for 3.2 as follows “rhMG53 treatment mitigates NM-induced epidermal structure damages”.

Reviewer 4 Report
Thank you for the opportunity to review this interesting paper. Hopefully the authors find my comments and suggestions helpful.
General comments:
-
There are quite a lot of references per sentence. For example, line 39 has seven references, line 43 has eight references, and line 45 has eighteen references. Try to be more specific and select key references throughout the paper. Also, list specific references next to different topics in one sentence. For example, The skin toxicity caused by mustard vesicants mainly reflects their alkylating properties and/or thiol-de- pleting functions, which lead to DNA damage (ref), free radical formation (ref), and lipid peroxidation(ref).
-
Many of the methods are provided in the results.
-
The results can be described in more detail.
-
Do these agents only target keratinocytes and hair follicle stem cells? Does it cause alopecia?
-
More information about MG53 knockout mice and associated pathology would be interesting, considering it is a key component of the present study.
-
Does the rejuvenation of hair follicle stem cell (HFSC) function result in wound healing or reduced scarring? How?
-
How were the dosages 3.2 and 0.8 selected?
-
The methods need to provide greater detail of the experiments conducted, especially NM exposure dosages, times, and treatments. As is, it can be not very clear.
-
Please provide more detail for Fig. 3b
-
Try to remove “we” from the article. For example: “In this study, we provide the data to show the potential therapeutic value of MG53 to treat mustard-250 induced skin injury.” can be changed to: This study shows the potential therapeutic value of MG53 to treat mustard-250 induced skin injury.
Specific comments:
Line 24: Period after repair > MG53 is a vital component of cell membrane repair. We have…
Line 63: wounding or wound healing?
Line 82: Period after NJ). Mice were exposed…
Line 83: NM concentration?
Line 86: Are mice treated on five consecutive days or only once, and measurements performed five days later?
Line 96: Add space after previously
Line 135: Is the “more severe response” limited to reduced activity, or were there other obvious or topical indicators?
Line 137: was the reduction significant? Was this quantified?
Line 156: blistering was not mentioned for the higher dosage.
Line 156: Methods section mentioned exposure for five days?
Line 158 - 159 change sentence to NM-induced skin injury was less severe in the tPA-MG53 mice (Fig. 2a).
Line 159: The authors make reference to 28 days post-exposure. As a result, the method section needs to be more detailed to have a better understanding of the experiment. Please update the method description with the appropriate exposure lengths etc. Perhaps in table format if appropriate.
Lines 181 - 183 is a description of the method, not a result. The result: visible differences in skin architecture could be described in greater detail. While it is mentioned that changes were quantified, the results are not provided.
Lines 181 - 184 is a description of the method, not a result.
Line 235-236: FACS analyses were used to quantify the changes in ROS levels in HFSCs treated with NM (plus saline) 235 and NM (plus rhMG53) (Fig. 4b). Fig 4b does not show ROS levels? How does DCF relate to ROS? Please explain Fig 4b
Line 252: Is tPA-MG53 mice the wildtype? It is not clear from sentence construction.
Line 258: change again to against
Overall the English are good. There is an unnecessary reference to "we" throughout the paper.
Author Response
Reviewer 4:
Comments and Suggestions for Authors
Thank you for the opportunity to review this interesting paper. Hopefully the authors find my comments and suggestions helpful.
General comments:
There are quite a lot of references per sentence. For example, line 39 has seven references, line 43 has eight references, and line 45 has eighteen references. Try to be more specific and select key references throughout the paper. Also, list specific references next to different topics in one sentence. For example, the skin toxicity caused by mustard vesicants mainly reflects their alkylating properties and/or thiol-depleting functions, which lead to DNA damage (ref), free radical formation (ref), and lipid peroxidation(ref).
We appreciate the reviewer’s suggestion, and have revised the reference citations as recommended.
Many of the methods are provided in the results. The results can be described in more detail.
We have followed the recommendations of all 4 reviewers and made revisions to more clearly present our findings. We appreciate all reviewers’ constructive recommendations.
Do these agents only target keratinocytes and hair follicle stem cells? Does it cause alopecia?
In this study, we focused on keratinocyte and hair follicle stem cells which contribute more on cutaneous wound healing based on our previous studies (and other studies as well). We shaved mice for NM exposure, and we did not have any data on alopecia. It will be interesting to study the hair loss (alopecia) in our future study.
More information about MG53 knockout mice and associated pathology would be interesting, considering it is a key component of the present study.
Please see our response to reviewer 1.
Does the rejuvenation of hair follicle stem cell (HFSC) function result in wound healing or reduced scarring? How?
We have added the following sentences to the Discussion section (see page 9, line 264-269):
“In a recent study[44], we showed that mice with ablation of MG53 display defective hair follicle structure, and topical application of rhMG53 can promote hair growth in the mg53−/− mice. We further find that rhMG53 protects HFSCs from oxidative stress-induced apoptosis and stimulates differentiation of HSFCs into keratinocytes[44]. Additionally, MG53 has been shown to play an anti-inflammatory role in viral infection, chronic injury and aging [62, 63]. All these findings link MG53’s function to the improved regenerative capacity to facilitate wound healing.”
How were the dosages 3.2 and 0.8 selected?
We followed the protocols developed by Jain et al (2014) and Tewari-Singh et al (2014).
The methods need to provide greater detail of the experiments conducted, especially NM exposure dosages, times, and treatments. As is, it can be not very clear.
We have revised the method section as recommended. Please also see our responses to reviewer 1 and 3.
Please provide more detail for Fig. 3b
Agree, more details are added to the figure legend for Fig. 3b. Please see line 325-327.
“The repair-patch remained stable over the 2 min observation period in control cells not ex-posed to NM (green). Cells treated with NM displayed dysfunctional GFP-MG53 movement following microelectrode induced membrane injury (red).”
Try to remove “we” from the article. For example: “In this study, we provide the data to show the potential therapeutic value of MG53 to treat mustard-250 induced skin injury.” can be changed to: This study shows the potential therapeutic value of MG53 to treat mustard-induced skin injury.
We have modified as recommended.
Specific comments:
Line 24: Period after repair > MG53 is a vital component of cell membrane repair. We have…
We have made corrections as recommended.
Line 63: wounding or wound healing?
We have made corrections as recommended.
Line 82: Period after NJ). Mice were exposed…
We have made corrections as recommended.
Line 83: NM concentration?
We used different doses of NM in our studies: high dose with 3.2mg/mouse was only used to study the systemic NM injury and low dose (2.0 and 0.8mg/mouse) was used for the acute and delay NM injury as described in our results section 3.1 and figure legends respectively.
Line 86: Are mice treated on five consecutive days or only once, and measurements performed five days later?
Mice were treated with rhMG53/cream (once) plus SC rhMG53 for a consecutive 3 days.
Line 96: Add space after previously.
We have made correction as recommended.
Line 135: Is the “more severe response” limited to reduced activity, or were there other obvious or topical indicators?
We mentioned the more severe response based on the observation of animal behavior as shown in Supplementary movies. Please see response to reviewer 1.
Line 137: was the reduction significant? Was this quantified?
We have changed the word from “remarkable” to “measurable”, on page 6, line 161.
Line 156; Blistering was not mentioned in higher dosage.
We have added the blistering in higher dosage injury.
Line 156: Methods section mentioned exposure for five days?
All mice were treated with one-time NM topical application. The five days reflect the total rhMG53 treatment.
Line 158 - 159 change sentence to NM-induced skin injury was less severe in the tPA-MG53 mice (Fig. 2a).
We have made change as recommended.
Line 159: The authors make reference to 28 days post-exposure. As a result, the method section needs to be more detailed to have a better understanding of the experiment. Please update the method description with the appropriate exposure lengths etc. Perhaps in table format if appropriate.
We have modified our method section as recommended. Please also see our responses to reviewer 1 and 3.
Lines 181 - 183 is a description of the method, not a result. The result: visible differences in skin architecture could be described in greater detail. While it is mentioned that changes were quantified, the results are not provided. Lines 181 - 184 is a description of the method, not a result.
The quantifiable results were presented in Fig. 2c.
Line 235-236: FACS analyses were used to quantify the changes in ROS levels in HFSCs treated with NM (plus saline) 235 and NM (plus rhMG53) (Fig. 4b). Fig 4b does not show ROS levels? How does DCF relate to ROS? Please explain Fig 4b
We used DCF as an additional fluorescent indicator for ROS, and have added the details in the revised manuscript. Please see line 220-225:
2’,7’-dichlorodihydrofluorescein diacetate (DCFH-DA,) staining has been used for quantification of ROS detection [57]. Oxidation of DCFH by ROS converts the molecule to DCF, which emits green fluorescence at an excitation wavelength of 485 nm and an emission wavelength of 530 nm. FACS analyses were thus used to quantify the changes in DCF levels in HFSCs treated with NM (+saline, pink) and NM (+rhMG53, green) (Fig. 4b). Clearly, the left shift in the peak distribution of DCF reflected the anti-oxidative stress function of rhMG53 in HFSCs following NM exposure.
Line 252: Is tPA-MG53 mice the wildtype? It is not clear from sentence construction.
tPA-MG53 mice is transgenic mice with sustained elevation of MG53 in the bloodstream. We have added the details and reference about the tPA-MG53 mouse in method section 2.2.
Line 258: change again to against
Sorry for the mistake. We have made correction as recommended.

Round 2
Reviewer 3 Report
The manuscript has been improved by author's response to reviewer comments. Two concerns were not adequately addressed and are restated below.
1. Figure 1:
Original Reviewer Comment: (Fig 1c and e) Since WT and mg53 -/- mice are different genotypes, it is necessary to analyze organ weight/length in untreated, age-matched animals to conclude the differences observed are due to NM treatment response and not, in part, due to normal physiological differences.
Author Response: We have analyzed all organ-weight in different genetic ground following NM exposure (2 mg/mouse). We used sham as untreated control. The significant differences were only observed in liver and spleen.
V2 Reviewer Comment: It is clear that the organs were analyzed following NM exposure. Unless the organs are analyzed prior to NM exposure (and show no significant differences), it is not possible to conclude that the observed differences are due to NM response and not due to baseline genotypic differences in organ size. Colon length, and spleen and liver weights should be measured in age-matched, unexposed WT and mg53 -/- mice in order to draw such conclusions.
2. Figure 2:
Fig. 2d (hair follicle quantitation for mg53 -/- experiment) should move to become Fig. 2b to coordinate with images from relevant experiment (in Fig. 2a). It remains unclear as to why hair follicles were not quantified for the rhMG53 treatment experiment (Fig. 2b,c) as well.
Author Response
Comments and Suggestions for Authors
The manuscript has been improved by author's response to reviewer comments. Two concerns were not adequately addressed and are restated below.
- Figure 1:
Original Reviewer Comment: (Fig 1c and e) Since WT and mg53 -/- mice are different genotypes, it is necessary to analyze organ weight/length in untreated, age-matched animals to conclude the differences observed are due to NM treatment response and not, in part, due to normal physiological differences.
Author Response: We have analyzed all organ-weight in different genetic ground following NM exposure (2 mg/mouse). We used sham as untreated control. The significant differences were only observed in liver and spleen.
V2 Reviewer Comment: It is clear that the organs were analyzed following NM exposure. Unless the organs are analyzed prior to NM exposure (and show no significant differences), it is not possible to conclude that the observed differences are due to NM response and not due to baseline genotypic differences in organ size. Colon length, and spleen and liver weights should be measured in age-matched, unexposed WT and mg53 -/- mice in order to draw such conclusions.
We appreciate the reviewer’s suggestion, and have added the following texts to our revised manuscript on page 6 (line 158-160):
“We examined the morphology changes in organ weight and intestinal length between mg53-/- and age-matched wild type mice, and did not observe any significant difference with those phenotypes. At day 5 post NM exposure, mice were sacrificed, and histological examinations were conducted with the major vital organs (including heart, lung, liver, kidney, spleen, and colon) (Fig. 1b and 1c)”.
- Figure 2:
Fig. 2d (hair follicle quantitation for mg53 -/- experiment) should move to become Fig. 2b to coordinate with images from relevant experiment (in Fig. 2a). It remains unclear as to why hair follicles were not quantified for the rhMG53 treatment experiment (Fig. 2b,c) as well.
Thanks for your suggestion and we have revised Fig.2 as recommended.
We also acknowledge the reviewer’s recommendation by adding the following sentence on page 7 (line 190-193):
“In a recent study [44], we showed that mice with ablation of MG53 display defective hair follicle structure, and topical application of rhMG53 promoted hair growth in the mg53−/− mice. It will be interesting to study more on hair follicle stem cell function and long-term effect of rhMG53 treatment including the hair follicle development following NM exposure in the future.”
Reviewer 4 Report
This is a very interesting study on the role of MG53 in nitrogen mustard-induced skin injury. Thank you to the authors for the revisions provided.
A few minor suggestions:
Line 38: Add “a”: “It is a readily available”
Line 40: Change “personals” to “personnel”: “and military personnel”
Line 47: Change” radical” to radicals: “generate free radicals”
Line 89: Change 8-10 to Eight to ten: Eight to ten-week-old male
Line 98: Add “the” and change “studying” to “study”: the systemic toxicity study
Line 250 Add “A”: “A similar finding”
Line 252: Change “has been” to “is”: “...staining is used”
Author Response
This is a very interesting study on the role of MG53 in nitrogen mustard-induced skin injury. Thank you to the authors for the revisions provided.
We appreciate the reviewer’s suggestions and have revised as recommended.
A few minor suggestions:
Line 38: Add “a”: “It is a readily available”
We have made corrections as recommended.
Line 40: Change “personals” to “personnel”: “and military personnel”
We have made corrections as recommended.
Line 47: Change” radical” to radicals: “generate free radicals”
We have made corrections as recommended.
Line 89: Change 8-10 to Eight to ten: Eight to ten-week-old male
We have made corrections as recommended.
Line 98: Add “the” and change “studying” to “study”: the systemic toxicity study
We have made corrections as recommended.
Line 250 Add “A”: “A similar finding”
We have made corrections as recommended.
Line 252: Change “has been” to “is”: “...staining is used”
We have made corrections as recommended.